# Clinical Efficacy of Melon GliSODin^®^ for the Treatment of Aging-Related Dysfunction in Motor Organs—A Double Blind, Randomized Placebo-Controlled Study

**DOI:** 10.3390/jcm11102747

**Published:** 2022-05-12

**Authors:** Masato Koike, Masashi Nagao, Yoshiyuki Iwase, Kazuo Kaneko, Muneaki Ishijima, Hidetoshi Nojiri

**Affiliations:** 1Department of Orthopaedic Surgery, Juntendo Tokyo Koto Geriatric Medical Center, Tokyo 136-0075, Japan; ortho-m.koike@juntendo.ac.jp (M.K.); iwase@juntendo.gmc.ac.jp (Y.I.); 2Department of Orthopaedic Surgery, Juntendo University Graduate School of Medicine, Tokyo 113-8421, Japan; nagao@juntendo.ac.jp (M.N.); k-kaneko@juntendo.ac.jp (K.K.); ishijima@juntendo.ac.jp (M.I.); 3Department of Orthopaedic Surgery, Medical Technology Innovation Center, Clinical Research & Trial Center, Graduate School of Health and Sports Science, Juntendo University, Tokyo 113-8421, Japan

**Keywords:** body fat percentage, locomotive syndrome, melon glisodin, osteoarthritis, osteoporosis, oxidative stress, sarcopenia, superoxide dismutase

## Abstract

Background: Locomotive syndrome is a concept proposed in Japan involving decreased mobility due to osteoarthritis, osteoporosis, and sarcopenia. This double-blind, randomized study aimed to investigate the effects of superoxide dismutase (SOD)-rich melon extract (Melon GliSODin^®^) on locomotive syndrome. Methods: For 6 months, we administered oral Melon GliSODin^®^ (500.4 mg/day) or a placebo to 24 and 22 women, respectively (aged 50–80 years), with knee or lower back discomfort or pain. Using baseline and 6-month data, changes in the Verbal Rating Scale and in subjective symptoms (determined using the Japanese Knee Osteoarthritis Measure, Locomo 25, the Roland–Morris Disability questionnaire, and the Chalder Fatigue Scale) were assessed, along with various oxidative markers, antioxidants, inflammatory markers, renal and liver function biochemical markers, bone metabolism markers, body composition, and motor function. Results: Oral Melon GliSODin^®^ administration tended to be associated with a larger improvement in subjective symptom scores, a reduction in oxidative markers (malondialdehyde and diacron reactive oxygen metabolites) and tumor necrosis factor-α, and a significant increase in non-fat mass between baseline and 6 months. However, no statistically significant differences were observed between the groups for outcomes at 6 months. Conclusions: Melon GliSODin^®^ tended to improve the subjective symptoms of participants who had knee or lower back pain or discomfort. Melon GliSODin^®^ administration may help to prevent the progression of locomotive syndrome. Future studies involving larger sample sizes and more stringent randomization protocols are needed to determine differences between the placebo and Melon GliSODin^®^ groups.

## 1. Introduction

Population aging is progressing rapidly in high-income countries that are experiencing growth in both the size and proportion of the older adult population. In 2019, there was an estimated global population of 703 million people aged ≥65 years, and this number has been projected to double to 1.5 billion by 2050 [1]. As aging progresses, motor function gradually declines, causing disability in a large number of people. Deterioration in motor organ function due to age-related muscle weakness, joint and spinal diseases, and osteoporosis has been termed “locomotive syndrome’’ (Locomo) in Japan [2]. 

Knee osteoarthritis (OA), a chronic progressive joint disease associated with cartilage degeneration and restricted physical activity due to joint pain, has also been reported to be a strong risk factor in Locomo [3]. The prevalence and severity of both knee OA [4] and Locomo [2,5] are higher in women. Several studies have shown an association between OA and oxidative stress [6,7,8,9]. Oxidative stress is defined as an increase in reactive oxygen species (ROS) in the body, due to either an increase in the production system or a decrease in the scavenging system (ROS-scavenging antioxidant enzymes) in the cells. ROS increases due to disruption of the redox balance, causing tissue damage in various medical conditions, including cancer, cardiovascular disease, degenerative diseases, and infectious diseases [10]. Downregulation of superoxide dismutase (SOD) in human knee OA cartilage [7,8,9] and *Sod2* loss in chondrocytes has been shown to accelerate cartilage degeneration in a murine surgical OA model [11]. Further, *Sod2* deficiency in osteocytes induces bone loss [12], and *Sod2* deficiency in skeletal muscle induces muscle fatigue [13]. These findings suggest a close association between Locomo and oxidative stress, and that controlling oxidative stress may be an important interventional approach to Locomo [11,12,13,14].

SOD-rich melon extract (Melon GliSODin^®^) is an oral supplement in which SOD extracted from melon is combined with wheat gliadin. Although Melon GliSODin^®^ has been shown to activate the antioxidant enzyme SOD [15] and has beneficial effects in various diseases [15], its effect on Locomo is unknown. This study aimed to determine the effects of oral Melon GliSODin^®^ administration in female patients with Locomo.

## 2. Materials and Methods

### 2.1. Study Design

This prospective, randomized, double-blind, placebo-controlled study was conducted at Juntendo Tokyo Koto Geriatric Medical Center, Tokyo, Japan. Our study was approved by the Juntendo Clinical Research Review Board (CRB3180012) and conducted in accordance with the principles of the Declaration of Helsinki. The supporting CONSORT checklist is available in the supporting information. The study was registered in the Japan Registry of Clinical Trials (jRCT) and the University Hospital Medical Information Network (UMIN) (jRCTs 031180310 and UMIN 000025163, respectively). Written informed consent was obtained from each participant prior to beginning the study. 

### 2.2. Participants

Participants who met all of the inclusion and exclusion criteria participated in the study. Inclusion criteria comprised: (i) females, aged 50–80 years, (ii) with low back or knee pain and gait instability due to motor weakness, (iii) who were non-smokers, and (iv) who had not consumed any nutritional supplements that affect oxidative stress up to one month prior to participation in the clinical study. Exclusion criteria comprised those who had: (i) any systemic diseases that interfere with motor function, (ii) any psychogenic disorders or neuromuscular diseases, (iii) continuing orthopedic treatments at other hospitals, (iv) a history of orthopedic surgery on the lower back or lower limb joints, or, (v) an allergy to wheat or melon. Eligible participants were registered and participated in the study from December 2016 to September 2020.

### 2.3. Intervention

Melon GliSODin^®^ is a melon extract that combines wheat gliadin and SOD extracted from Vauclusien melon, which is cultivated in Avignon, in southern France. Melon GliSODin^®^ was manufactured by ISOCELL, NUTRA S.A.S (Paris, France) and imported and provided by Nutrition Act Co., Ltd. (Tokyo, Japan).

Participants were enrolled by M.T. and allocated to either an intervention or a placebo group through simple randomization using a computer-generated randomization table. Both participants and investigators were blinded to the study groups. Participants were prescribed Melon GliSODin^®^ (melon extract 83.4 mg, dextrin 64.6 mg, calcium stearate 2 mg per capsule, 500.4 mg/day, 6 capsules orally before breakfast) or a placebo (dextrin 118.2 mg, calcium stearate 2 mg per capsule, 6 capsules orally before breakfast) provided by Nutrition Act, Co., Ltd. (Tokyo, Japan). All the agents were given at the beginning of the study and participant follow-ups occurred every 1 or 2 months, as scheduled, for 6 months.

### 2.4. Outcomes

The following outcomes were assessed at baseline and at 6 months. We also evaluated the presence of gastrointestinal symptoms to assess the adverse effects of oral supplementation in the study participants.

### 2.5. Subjective Symptoms 

Subjective symptoms were assessed as the primary outcomes of the study. The severity of Locomo was classified as pre-, grade 1, and grade 2, using the Locomo 25 questionnaire and stand-up or two-step tests according to published criteria [2,16]. Quality of life associated with knee OA function was assessed using the Japanese Knee Osteoarthritis Measure (JKOM) [17], which is a patient-based, self-completed evaluation of OA comprised of four subcategories: pain and stiffness, activities of daily living, social activities, and general health condition. The degree of physical fatigue was assessed using the Chalder Fatigue Scale [18]. The degree of pain was assessed using the Verbal Rating Scale (VRS) for pain, with scores ranging from 0 (no pain) to 5 (very severe pain), as previously described [19]. Quality of life associated with lumbar OA was assessed using the Roland–Morris Disability questionnaire (RDQ), as previously described [20]. The RDQ questionnaire consists of 24 items and aims to determine the degree of impairment due to lower back pain in everyday activities, such as standing, walking, dressing, and working.

#### 2.5.1. Oxidative Markers, Antioxidants, and Inflammatory Markers

Serum levels of pentosidine, malondialdehyde (MDA), serum SOD activity, glutathione peroxidase (GPx), and tumor necrosis factor-α (TNFα) were measured at the Japan Institute for the Control of Aging and at NIKKEN SEIL Co., Ltd. (Shizuoka, Japan). Diacron reactive oxygen metabolites (dROMs) and biological antioxidant potential (BAP) were evaluated using a free radical analyzer (FREE Carrio Duo, Diacron International SRL, Grosseto, Italy). High-sensitivity C-reactive protein (hsCRP) and interleukin-6 (IL-6) were measured by SRL, Inc. (Tokyo, Japan).

#### 2.5.2. Biochemical Tests, Body Composition, and Motor Function

Biochemical tests were performed at baseline and 6 months after participation. Serum total protein (TP), albumin (Alb), aspartate aminotransferase (AST), alanine aminotransferase (ALT), blood urea nitrogen (BUN), creatinine (Cre), calcium (Ca), and phosphorus (Pi) were measured. Type I procollagen-N-propeptide (P1NP), tartrate-resistant acid phosphatase 5b (TRACP5b), osteocalcin (OC), and undercarboxylated osteocalcin (ucOC) were measured by SRL, Inc. (Tokyo, Japan). Body fat percentage, fat mass, non-fat mass, and bone density were measured using a prodigy dual X-ray absorptiometry system (GE Healthcare Japan Co., Ltd., Tokyo, Japan). Motor function was evaluated according to a previous study [2]. Briefly, a Smedley-type analog grip dynamometer (Takei Kikai Kogyo T.K.K.5001 Grip-A, Takei Scientific Instruments Co., Ltd., Niigata, Japan) was used to measure grip strength. Two measurements were performed alternately on the left and right hands, and the higher value for each was recorded. The mean value for the left and right hands was then rounded to two decimal places. Lower extremity extension torque was measured using Strength Ergo240 (Mitsubishi Electric Engineering Co., Ltd. Tokyo, Japan), as previously described [21]. The 6-minute walking distance test, two-step tests, and TUG measurements were performed as previously described [22,23,24]. The ability to stand up with a single- or double-leg stance from stools at heights of 40, 30, 20, and 10 cm was evaluated, as previously described [2,25]. 

### 2.6. Sample Size Analysis

Based on a previous report [26] in which the SOD concentration of erythrocytes was significantly higher following oral Melon GliSODin^®^ supplementation (1974 ± 186 UI/gHb, *n* = 10) compared with placebo (1769 ± 223 UI/gHb, *n* = 9), we estimated a sample size of 21 participants would detect a mean difference of 205 UI/gHb, at a standard deviation of 200 UI/gHb, at 5% on both sides, and with a detection power of 90%. We also estimated that approximately 20% of the participants in each group would drop out of the study; therefore, the sample size was set to 25 per group, or 50 in total.

### 2.7. Statistical Analysis

Statistical analyses were conducted using SPSS 26.0 software (IBM, Inc., Armonk, NY, USA). After testing the data for normality, Student’s *t*-tests or Mann–Whitney U tests were used to compare the 2 groups, as appropriate, at baseline and at 6 months, as well as in terms of baseline-to-6-month changes in values. An analysis of covariance (ANCOVA) was performed to compare 6-month outcomes between the groups, using the baseline (0 month) value of each item as a covariate and the 6-month value as the dependent variable. A two-sided level of 5% was considered statistically significant.

## 3. Results

A CONSORT 2010 flow diagram of the study is shown in Figure 1. In total, fifty female participants were enrolled in the study, and one participant declined to participate after enrollment. Forty-nine participants were randomized: 25 to the Melon GliSODin^®^ group and 24 to the placebo group. One participant in the Melon GliSODin^®^ group and two participants in the placebo group were lost to follow-up between baseline and the 2-month appointments. The remaining 46 participants (Melon GliSODin^®^ group, *n* = 24, placebo group, *n* = 22) adhered to the Melon GliSODin^®^/placebo regimen, completed follow-up, and were analyzed for the study by per-protocol analysis (Figure 1).

The baseline characteristics of the 46 participants are shown in Table 1. There were no significant differences in baseline characteristics between the three participants who were lost to follow-up and the 46 who completed follow-up (data not shown), allaying concerns of selection bias in per-protocol analysis. There were no significant differences in age, height, body weight, body mass index (BMI), Locomo grade, Locomo 25, and RDQ between the two groups. Concerning participants in both groups, 50% in the placebo group and 54% in the Melon GliSODin^®^ group were found to have grade 2 Locomo. However, JKOM, the Chalder Fatigue Scale, and the VRS pain scores were significantly higher in the Melon GliSODin^®^ group than in the placebo group. No participants had any noticeable gastrointestinal symptoms.

There were no significant differences between the 2 groups in terms of baseline values concerning oxidative markers, antioxidants, and inflammatory markers, blood biochemistry, bone metabolism, body composition, and motor function (with the exception of the stand-up test score, which was significantly higher in the placebo group, Appendix A).

## 4. Effects of Melon GliSODin^®^ on Symptom Severity

There were no significant differences in Locomo 25, JKOM, Chalder Fatigue Scale, and VRS and RDQ scores between the 2 groups at 6 months. The decrease in scores from baseline to 6 months tended to be larger in the Melon GliSODin^®^ group than in the placebo group; however, no statistical differences were observed (Table 2).

## 5. Effects of Melon GliSODin^®^ on Oxidative Markers, Antioxidants, and Inflammatory Markers 

There were no significant differences in circulating concentrations of oxidative markers (plasma pentosidine, MDA, and dROMs), major antioxidants (serum SOD activity, serum GPx, and BAP), or inflammatory markers (TNFα, hsCRP, and IL-6) between the two groups at 6 months. (Appendix A) Furthermore, the 6-month values did not significantly differ between the two groups after adjusting for baseline values (ANCOVA, results not shown), or when comparing changes from baseline to 6 months (Appendix A); however, there was a tendency towards reduction in oxidative markers (MDA and dROMs) and TNFα in the Melon GliSODin^®^ group.

## 6. Effect of Melon GliSODin^®^ on Blood Biochemistry, Bone Metabolism, Body Composition, and Motor Function 

There were no statistically significant differences in the levels of biochemical markers for renal and liver function, bone metabolism markers, body composition, and motor function (with the exception of the stand-up test scores) between the 2 groups at 6 months (Appendix A). A comparison of baseline-to-6-month changes in values showed no significant differences in all the above outcomes, with the exception of non-fat mass, which showed a significant increase in the Melon GliSODin^®^ group compared with the placebo group (Appendix A).

## 7. Discussion

In this study, the subjective symptom scores for Locomo 25, JKOM, VRS, RDQ, and the Chalder Fatigue Scale tended to improve over 6 months in the Melon GliSODin^®^ group. There was a tendency for reduction in oxidative markers (MDA and dROMs) and TNFα, and a significant increase in non-fat mass in the Melon GliSODin^®^ group compared with the placebo group.

Malondialdehyde (MDA) is one of the final products of polyunsaturated fatty acids peroxidation in cells and is a marker of oxidative stress, and MDA levels have been reported to increase in patients with knee OA [27,28]. Other common biomarkers of oxidative stress are dROMs, which are used as an index of the products from ROS [29]. TNFα is an inflammatory cytokine with various functions, and a recent study has reported its association with pain [30]. Serum TNFα, along with IL-6, is known to be associated with spontaneous pain in OA, knee pain, and lower back pain [31]. In our study, although serum TNFα reduced in the Melon GliSODin^®^ group, no differences were observed in terms of patient-reported outcome measures such as JKOM. Therefore, although Melon GliSODin^®^ had an antioxidant and anti-inflammatory effect in the patients with Locomo, we did not observe any significant improvement in knee and lower back pain.

We measured serum SOD activity throughout the study; however, we did not observe any significant changes over time in both groups. Evidence on the effect of Melon GliSODin^®^ on serum SOD levels has been controversial. In a study involving healthy individuals who were administered GliSODin for 14 days, oxidative stress loading was performed in a high-pressure oxygen environment, and DNA damage was evaluated using a comet assay [32]. In the Melon GliSODin^®^ groups, DNA damage due to oxidative stress loading was suppressed, but blood SOD levels did not change [32]. However, a correlation between decreased serum SOD activity and prognosis in older adult patients with gastric cancer has been reported [33]. We hypothesized that Melon GliSODin^®^ administration could reduce SOD activity. However, given no differences were observed between the placebo and Melon GliSODin^®^ groups over time, serum SOD levels may not represent body SOD activity in patients with pain due to Locomo.

This study had several strengths and limitations. One strength of this study was its randomized, controlled, double-blinded, prospective interventional design. Participants were randomly assigned, and the assignment was concealed. Additionally, the study had complete (100%) adherence rate, with a follow-up rate of 90% throughout the study period. However, our study had some limitations. First, although there were no statistically significant differences in the participants′ baseline characteristics in terms of height, weight, BMI and Locomo grade, there were significant differences in JKOM and Chalder Fatigue scores, with patients with more severe pain in the Melon GliSODin^®^ group. We controlled for confounding from different baseline values using ANCOVA analysis; however, differences at baseline may have affected the results. Second, the sample size estimation prior to the study was inaccurate. We calculated the sample size based on a previous study that had used erythrocyte SOD concentrations to assess the effect of Melon GliSODin^®^ [34]. In our study, we assessed serum SOD levels and observed a much higher variation than anticipated in terms of the previous report, thereby underpowering our analyses. Third, only female participants were enrolled in the study, and the results may not be directly extrapolated to male patients with Locomo. Further studies using larger study samples, including male participants and more stringent randomization protocols, should be considered.

In conclusion, Melon GliSODin^®^ tended to improve the subjective symptoms of participants who had knee or lower back pain or discomfort. Melon GliSODin^®^ administration may help to prevent the progression of Locomo. Further well-designed clinical studies are needed to confirm the effects of Melon GliSODin^®^.

## Figures and Tables

**Figure 1 jcm-11-02747-f001:**
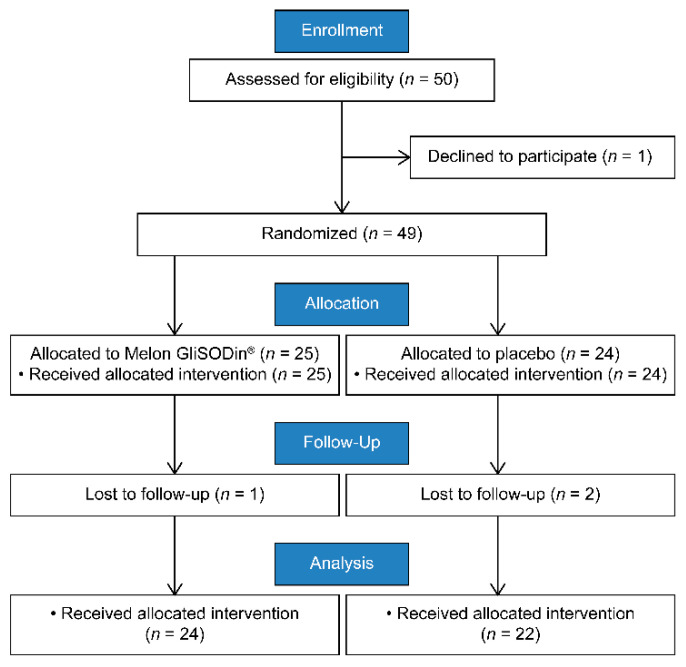
Flowchart outlining the study design.

**Table 1 jcm-11-02747-t001:** Study participants’ baseline characteristics in the Melon GliSODin^®^ (*n* = 24) and placebo (*n* = 22) groups.

	Melon GliSODin^®^	Placebo	*p*-Value
Age	68.25 (6.96)	65.95 (7.83)	0.30
Height, cm	154.16 (5.05)	153.63 (5.18)	0.71
Body weight, kg	54.22 (7.48)	53.33 (5.47)	0.96 ^#^
BMI, kg/m^2^	22.40 (2.32)	22.70 (3.15)	0.84 ^#^
Locomo Grade, *n*			
Pre-Locomo, *n*	1	6	
Grade 1, *n*	10	5	
Grade 2, *n*	13	11	
Locomo 25	15.63 (11.89)	10.82 (12.11)	0.06 ^#^
JKOM	46.71 (13.37)	40.18 (13.03)	0.04 ^#,^*
Chalder Fatigue Scale	18.75 (7.28)	14.55 (6.70)	0.05 *
VRS	2.75 (0.94)	2.23 (1.15)	0.05 ^#^
RDQ	4.33 (4.17)	3.00 (4.05)	0.21 ^#^

Values are shown as mean (SD), ^#^
*p*-values obtained using Mann–Whitney U tests, all others using Student’s *t*-tests, * *p* < 0.05. Abbreviations: JKOM, Japanese Knee Osteoarthritis Measure; RDQ, Roland–Morris Disability questionnaire; VRS, Verbal Rating Scale.

**Table 2 jcm-11-02747-t002:** The effects of Melon GliSODin^®^ on symptom severity comparison between the GliSODin (*n* = 24) and placebo (*n* = 22) groups after 6 months.

Outcomes	6-Month Outcome			Baseline to 6-Month Change		
	Melon GliSODin^®^	Placebo	*p*-Value	Melon GliSODin^®^	Placebo	*p*-Value
Locomo 25	13.38 (10.2)	11.91 (16.08)	0.39	−2.25 (9.35)	1.09 (7.25)	0.19
JKOM	41.42 (13.16)	39.85 (16.57)	0.49	−5.29 (12.53)	−0.32 (10.01)	0.15
Chalder Fatigue Scale	15.54 (6.30)	13.95 (7.72)	0.45	−3.21 (7.02)	−0.59 (4.63)	0.15
VRS	2.33 (1.05)	2.14 (1.17)	0.75	−0.42 (1.25)	−0.09 (0.87)	0.31
RDQ	2.67 (2.65)	2.32 (4.11)	0.31	−1.67 (3.40)	−0.68 (2.61)	0.28

Values are shown as mean (SD). Abbreviations: JKOM, Japanese Knee Osteoarthritis Measure; RDQ, Roland–Morris Disability questionnaire; VRS, Verbal Rating Scale.

## Data Availability

The data presented in this study are available on request from the corresponding authors. The data are not publicly available due to privacy and ethical concerns.

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
