# Peer review of "Clinical Efficacy of Melon GliSODin® for the Treatment of Aging-Related Dysfunction in Motor Organs—A Double Blind, Randomized Placebo-Controlled Study"

_jcm, 2022, doi:10.3390/jcm11102747_

Round 1

Reviewer 1 Report

The study seems to have been well conducted, however, it can be reported more comprehensively.

There were some significant baseline imbalances, however, this has been handled appropriately, that is, by calculating the differences in change from baseline, and the baseline values were included in the ANCOVA. Furthermore, the drop-out rate was low.

Were the results analyzed using an intention-to-treat approach?

A lot of outcomes were analyzed – was a correction of the significance levels done? For example Bonferroni or Šidák correction?

Please consider adding confidence intervals to table 2.

I could not find the CONSORT checklist.

Please use headings in the abstract (background, methods, results, conclusion)

In the results section of the abstract, please report the results more comprehensively and indicate what was significant.

In the conclusion of the abstract, please make an interpretation of your findings and not just the limitations of the study.

Author Response

Response letter attached.

Reviewer 2 Report

The study title is focused on the Clinical Efficacy of Melon-GliSODin® for the Treatment of Aging-related Dysfunction in Motor Organs. Study is designed and conducted properly. As for the preliminary study it is enough. My suggestion is to plan study on the larger population not only on women.

Author Response

Response to Reviewer 2 Comments

Point 1: The study title is focused on the Clinical Efficacy of Melon-GliSODin® for the Treatment of Aging-related Dysfunction in Motor Organs. Study is designed and conducted properly. As for the preliminary study it is enough. My suggestion is to plan study on the larger population not only on women.

Response 1: Thank you for your kind and encouraging review. We agree with the reviewer’s suggestion and acknowledge the shortcoming of the current study being conducted only in small population of women. We included only women in this study as both knee OA and Locomo are not only more prevalent but also more severe in women. We have revised the Introduction and Methods section to clarify our rationale, and also the Discussion section to clarify this limitation and scope for future work.
